# Persistent Inflammation in Cerebral Palsy: Pathogenic Mediator or Comorbidity? A Scoping Review

**DOI:** 10.3390/jcm11247368

**Published:** 2022-12-12

**Authors:** Madison C. B. Paton, Megan Finch-Edmondson, Russell C. Dale, Michael C. Fahey, Claudia A. Nold-Petry, Marcel F. Nold, Alexandra R. Griffin, Iona Novak

**Affiliations:** 1Cerebral Palsy Alliance Research Institute, Speciality of Child and Adolescent Health, Sydney Medical School, Faculty of Medicine and Health, The University of Sydney, Sydney, NSW 2050, Australia; 2Children’s Hospital at Westmead Clinical School, Faculty of Medicine and Health, Sydney Medical School, University of Sydney, Sydney, NSW 2145, Australia; 3Kids Neuroscience Centre and T Y Nelson Department of Neurology and Neurosurgery, Children’s Hospital at Westmead, Sydney, NSW 2145, Australia; 4Department of Paediatrics, Monash University, Melbourne, VIC 3168, Australia; 5The Ritchie Centre, Hudson Institute of Medical Research, Melbourne, VIC 3168, Australia; 6Monash Newborn, Monash Children’s Hospital, Melbourne, VIC 3168, Australia; 7Faculty of Medicine and Health, The University of Sydney, Sydney, NSW 2050, Australia

**Keywords:** inflammation, biomarker, cerebral palsy, comorbidity, scoping review

## Abstract

Research has established inflammation in the pathogenesis of brain injury and the risk of developing cerebral palsy (CP). However, it is unclear if inflammation is solely pathogenic and primarily contributes to the acute phase of injury, or if inflammation persists with consequence in CP and may therefore be considered a comorbidity. We conducted a scoping review to identify studies that analyzed inflammatory biomarkers in CP and discuss the role of inflammation in the pathogenesis of CP and/or as a comorbidity. Twelve included studies reported a range of analytes, methods and biomarkers, including indicators of inflammatory status, immune function and genetic changes. The majority of controlled studies concluded that one or more systemic biomarkers of inflammation were significantly different in CP versus controls; most commonly serum or plasma cytokines such as tumor necrosis factor, Interleukin (IL)-6 and IL-10. In addition, differences in inflammation were noted in distinct subgroups of CP (e.g., those with varying severity). The available evidence supports the pathogenic role of inflammation and its ongoing role as a comorbidity of CP. This review shows that inflammation may persist for decades, driving functional impairment across development and into adulthood. However, inflammation is complex, thus further research will increase our understanding.

## 1. Introduction

Cerebral palsy (CP) describes a group of permanent motor and postural disorders, causing activity limitations, that are attributed to non-progressive disturbances in the developing brain [1]. The motor disorders of CP are commonly accompanied by sensation, perception, cognition, behavior and communication disturbances, with epilepsy and secondary musculoskeletal issues. Inflammation is detailed in the pathogenesis of most perinatal brain injury that contributes to the risk of developing CP, including neonatal stroke, preterm birth, birth asphyxia and infection [2,3]. Systematic reviews of the clinical literature now support an association between higher circulating levels of pro-inflammatory mediators and the diagnosis of CP, particularly in the setting of prematurity [4]. However, the duration and extent of this inflammation, as well as the implications in people with CP, remain unclear.

Inflammation in CP may comprise changes in cytokines [4]; altered immune response with a dysregulated response to stimuli (e.g., lipopolysaccharide (LPS)) [5]; adaptive immune changes including T- and B- cell distribution and function [6], and other genetic and non-genetic changes to signaling pathways [7]. It is commonly reported that inflammation following perinatal brain injury changes over time, between the acute to chronic phases of injury [8]. However, persistent inflammation (i.e., inflammation extending months-to-years after the primary injury phase) has been postulated to have detrimental effects on the brain and may contribute to ongoing sequelae of CP [9,10]. This is supported by the *sustained inflammation hypothesis,* also known as programming effects, whereby prenatal, antenatal or neonatal pro-inflammatory cytokines induce inflammation that contributes to long-term cytokine dysregulation [11]. Whilst it has been discussed that persistent inflammation may be present in people with CP [5,7], there is ongoing debate about the strength of this evidence and its implications. There are currently two main streams of thought: (1)*Inflammation persists well after the original brain injury with long-term consequence in CP.* If inflammation persists, other commonly characterized symptoms of CP, including pain and cognition may be confounded or exacerbated. For example, systemic immune disturbances in innate and cellular immunity increase brain glial cell responsiveness, which may worsen neurological deficits [12]. In this context, inflammation may be viewed as a contributing factor to the symptoms experienced by people with CP, and present with the motor and movement impairments, in other terms be a “comorbidity” of CP.(2)*Inflammation persists in CP without long-term consequence or indeed, does not persist at all.* If inflammation is not a comorbidity of CP, it may instead standalone as a pathogenic feature of CP; meaning that the original inflammatory insult that causes the motor and postural impairment exists with no other long-term implications on health or outcomes. Alternatively, that inflammation does not persist.

When considering either stream of thought, there are many unknowns around how and why inflammation may or may not persist. Crosstalk between inflammation arising from the central nervous and systemic immune system are also unclear, along with the possible contribution of epigenetic programming from the initial insult [13].

Interestingly, the role of inflammation in the pathogenesis of CP is frequently investigated; many publications report on inflammatory biomarkers either collected close to the time of birth (e.g., umbilical cord blood) or during the neonatal period up to 4 weeks of age, or study a mixed at-risk population which are then correlated with outcomes such as CP [14,15]. Early inflammatory biomarkers and associations with neurodevelopmental outcome can be seen as indirect measures that allude to the impact of inflammation, genetic and immune changes in the pathogenesis of brain injury. However, direct measures of inflammation in children and adults with CP is lacking and the extent and duration of inflammation is not understood. The need for more research in this space has been previously stated [7].

Hence, this scoping review aims to identify and synthesize results from clinical studies that analyze biomarkers of inflammation in established CP and discuss the role of inflammation in the pathogenesis of CP and as a comorbidity. In addition, as CP is highly heterogenous, we also aim to elucidate any effects within subgroups of CP, including etiology, age and type and topography of CP. Adequate identification of those with higher inflammation may open the potential for personalized medicine and targeted therapeutics, as well as identify responders/non-responders to treatments with an inflammation-modulating mechanism of action. 

## 2. Materials and Methods

We followed the Joanna-Briggs Institute *Population*, *Concept*, *and Context* keywords search method to formulate our scoping review question [16]. The study protocol was published on the Open Science Framework, July 2022 (DOI 10.17605/OSF.IO/6ZVUN (accessed on 17 November 2022)). This scoping review is reported in accordance with the *Preferred Reporting Items for Systematic Reviews and Meta-Analyses extension for Scoping Reviews Guidelines (PRISMA-ScR,* [17]). The PRISMA-ScR Checklist can be found in Appendix A. 

### 2.1. Study Eligibility Criteria

We included all study types (controlled and non-controlled) that reported on participants with CP. Studies were required to include the analysis of biomarkers that measured changes in inflammation, or markers of immune or inflammatory pathways, including genetic. Full text records, published in English in a peer-reviewed journal were eligible for inclusion. No limits were placed on the date of publication. 

We excluded studies that measured inflammation in infant populations prior to a CP diagnosis; captured biomarkers only for the purpose of analysis of risk factors susceptibility, associations and causal pathways to CP; or used neuroimaging, neuropathology or indirect functional measures as a biomarker.

### 2.2. Search Strategy

We ran searches on 28 June 2022 using MEDLINE (1946 to 28 June 2022), Cochrane Central (The Cochrane Library, June 2022) and Embase (1947 to 28 June 2022) via Ovid using the following strategy: (Cerebral palsy.tw) AND (Inflamm*.tw). Searches were limited to English language articles and de-duplicated.

### 2.3. Study Selection

De-duplicated search results were exported into Covidence Systematic Review Software (Veritas Health Innovation, Melbourne, Australia, available at: http://www.covidence.org (accessed on 17 November 2022)). Additional de-duplication was conducted before titles and abstracts were screened by two independent reviewers (MCBP and MFE). Full texts of studies were then retrieved and independently assessed for eligibility.

### 2.4. Data Extraction

Data was extracted by two independent review authors (MCBP and MFE) into a Microsoft Excel spreadsheet which was developed specifically for this review. Extracted information included study details, participants and groups, method of biomarker analysis, analyte details, significant study findings including subgroup analyses (comprising severity, type/topography, age and etiology). 

### 2.5. Defining Reportable Inflammatory Biomarkers

Consistent with the inclusion criteria and scope of this review, only biomarkers related to inflammation were included. We defined biomarkers as molecules, proteins (including cytokines and chemokines, as well as their receptors) or cells and characteristics of cellular function. This also included quantitative PCR as a biomarker of the amount of cytokine expression, at the RNA level. Whilst we did not limit on the type of analytes, we do not report on growth factors, hormones or neuropeptides. 

### 2.6. Data Synthesis

Results are reported as a summary of included studies and in relation to three biomarker result types of (1) cytokine analysis, (2) immune function and/or (3) genetic changes and gene expression. Briefly, we define immune function to mean changes in immune or blood cells upon stimulation; changes in cytokine analysis is defined by the differences in unstimulated expression of inflammatory markers including quantitative PCR at the RNA level; and, genetic changes and gene expression refer to an alteration or variation (including polymorphisms) in nucleotide sequences and DNA expression. Results of findings within CP subgroups related to inflammation and age, CP type, severity, topography and etiology are also synthesized, reported and discussed.

## 3. Results

### 3.1. Search Results

A total of 1489 records were identified following the search procedure shown in Figure 1. After 556 duplicates were removed, 933 studies were screened by title and abstract with 895 studies subsequently excluded. Following full-text screening for eligibility, 12 studies were included in this review [5,6,9,18,19,20,21,22,23,24,25,26].

### 3.2. Study Characteristics

Studies that had reportable information on inflammatory biomarkers indicating directionality of inflammation and immune responses compared to a control were synthesized and presented (Table 1) [5,6,9,18,19,21,22,24,25,26]. The two studies without a control group could not be summarized and reported in Table 1 [20,23]. However, their study characteristics and outcome data are presented below (see Section 3.4*. Additional findings in single arm studies*) and in Table 2, which describes sub-group differences. 

#### 3.2.1. Study Design

Of the 12 included studies, ten were controlled (83%) and two were single arm (17%). All controlled studies compared a CP group to those without CP, however features of the control group were highly variable and many studies had a number of comparator groups. Comparator groups included age-matched controls [6,9,19,24,27], gestational-age matched controls [5], infants, children, adolescents and adults with/without CP [6,18,19,21,22,25,26].

#### 3.2.2. Participant Features

Within the CP group, age was highly variable: one study recruited both infants (<1 year) and children (1–9 years) [25], three studies recruited children [19,21,24], seven studies recruited children and adolescents (10–19 years, [5,6,9,20,22,23,26] and one analyzed adolescents and adults (20–39 years, [18]). Included participants had variable type, topography, severity and etiology of CP (Table 1). 

#### 3.2.3. CP Cohort Etiology

Information regarding CP etiology was detailed in five included studies, with information provided in Table 1. One study recruited participants with asphyxia and/or infection [19], one recruited participants with preterm birth and PVL [5], one recruited participants with neonatal encephalopathy [6], and two had mixed participant etiologies (periventricular leukomalacia, birth asphyxia, hypoxia ischemic encephalopathy, neonatal encephalopathy, infection or stroke) [9,25]. All other studies did not provide information relating to CP etiology. A number of studies did specify participant gestational age at birth as a participant demographic. Whilst earlier gestational age (particularly very/extremely preterm) may be a risk factor for CP, it’s direct link to CP etiology was not specified. One study did investigate the contribution of gestational age on CSF inflammatory markers in a subgroup analysis [20]. In this study, preterm birth was presented as a CP etiology and details of these findings can be found in Section 3.5.4 below.

#### 3.2.4. Samples Analyzed, Method and Biomarker Details

A range of samples were studied for biomarkers of inflammation including plasma [18,22,24,25,28], serum [6,19], whole blood (including DNA) [6,9,23,25], peripheral blood mononuclear cells (PBMCs) [21], PBMC supernatant or RNA [5], cerebrospinal fluid (CSF) [20] and muscle [26]. Overall, most studies analyzed systemic inflammation from peripheral blood origins (Table 1). 

A total of 30 inflammatory analytes were assessed including a variety of cytokines and their receptors. The most common analytes included cytokines such as tumor necrosis factor (TNF) (n = 8 studies), interleukin (IL)-6 (n = 7 studies) and IL-10 (n = 5 studies). Other cytokines, including but not limited to IL-1β, transforming growth factor (TGF)-β1 and IL-8, were less commonly reported, only appearing in three or fewer studies. In addition, a variety of genes were investigated as well as lymphocyte populations (Table 1).

The most common method of analyte analysis was enzyme-linked immunosorbent assay (ELISA, n = 8 studies). Other methods included flow cytometry (n = 3 studies), reverse transcription-polymerase chain reaction (RT-PCR, n = 2 studies), complete blood counts with differential pictures of white blood cells (n = 1 study) and genotyping of single nucleotide polymorphisms (SNPs, n = 1 study). 

All studies had results that could be categorized as either biomarkers for (1) immune function [5,9,21], (2) inflammatory status via cytokine analysis [5,6,9,18,19,22,23,24,25,29] or (3) genetic changes and gene expression [5,25,26].

### 3.3. Results of Inflammatory Biomarkers in CP

Of the studies with a comparator group and presented in Table 1, n = 9/10 reported significant changes in inflammation (either inflammatory status, gene expression and/or immune response) in one or more findings in CP compared to a relevant comparator. These changes in inflammation were all systemic, except for one study assessing skeletal muscle [26]. It was noted that there was significantly higher systemic inflammation in adolescents with CP compared to relevant controls across four studies [5,6,9,22]. In the study measuring gene expression in skeletal muscle, higher inflammation was found in participants aged up to 18 years [26], with IL-1β, IL-6 and TNF mRNA being significantly higher in CP than controls. 

#### 3.3.1. Stimulation Assays of Immune Function from Case-Control Studies

Three studies analyzed systemic immune responses using stimulation of whole blood or PBMCs with LPS, a Toll-like receptor 4 agonist with an important role in regulation of immune responses to infection, or phorbol 12-myristate 13-acetate (PMA) and ionomycin [5,9,21] to induce cytokine responses in T cells. All studies found a significant increase in a range of cytokines in individuals with CP compared to controls. Statistically significant increases in cytokines for each study included TNF [5], granulocyte-colony stimulating factor (GM-CSF), IL-8 [21], and IL-1α, IL-2, IL-6 [9]. However, regulation of IL-1β was variable, with one study [9] reporting a significant increase following simulation in CP versus age-matched controls, whereas another study [21] found that IL-1β was significantly reduced in children with CP compared to adult controls. Together, these findings demonstrate that systemic immune response, whilst variable, is altered in participants with CP compared to individuals without CP.

#### 3.3.2. Cross-Sectional Cytokine Analysis from Case-Control Studies

A total of ten studies reported on the inflammatory status in CP; eight studies used serum and plasma to report on systemic inflammation [5,6,9,18,19,22,24,25], one analyzed CSF [20] and one analyzed skeletal muscle [26]. Overall there was significant heterogeneity in sample readouts measured via ELISA. TNF, IL-6 and IL-10 were most commonly reported and thus are discussed below for case–control studies. Plasma and serum ELISA results from three studies indicated significant increases in TNF in CP compared to a range of controls without CP [5,19,24]. One study reported no differences in serum/plasma TNF levels [9]. IL-6 was reported as being significantly increased in CP compared to controls in one study [19]. However, overall, IL-6 levels were reported to be not significantly different in CP versus controls in four other studies [5,6,9,22]. ELISA analysis of IL-10 indicated no significant differences in two studies [6,9] and significant increase in expression in one study of CP versus controls [25]. All other ELISA analytes varied significantly in frequency of reporting/analyzing and the outcome of the results (i.e., significant and non-significant findings, Table 1). Overall, whilst there is mounting data for differing TNF, IL-6 and IL-10 cytokine levels in CP versus controls, directionality was mixed. 

Differential mRNA expression between individuals with CP and their corresponding controls was investigated in two studies [5,26]. Significant increases in *IL-1β*, *IL-6*, *TNF* mRNA expression in skeletal muscle biopsies [26], and >2-fold increases in 10 TNF-related genes were found from unstimulated PBMCs in CP compared to controls [5]. Additionally, two studies reported two results that are not included in Table 1. In Lin et al. [5], RNA from PBMCs was analyzed in RT-PCR alongside protein expression via ELISA. This validated that *TLR-4,* inhibitor of nuclear factor kappa-B kinase subunit gamma, c-Jun N-terminal kinases and *TNF* genes were significantly higher in the CP group compared to controls. 

#### 3.3.3. Proportions of Immune Cell Types in Case-Controls

One study investigating cellular immunity from whole blood stimulated ex vivo with PMA and ionomycin or relevant cytokine ligand followed by flow cytometry analysis [6], indicated that T-cells of the adaptive and innate immune system (including overall, Vδ2, and CD4− CD8− subtypes) and invariant natural killer T (iNKT) cell percentage frequency and absolute numbers were elevated in school-aged children with CP compared to age-matched controls. Vδ1 T cells, Mucosal-associated invariant T (MAIT) cells and B-cells were significantly reduced in children with CP compared to age-matched controls. These results point to an altered distribution and frequency of adaptive immune cells and innate lymphocytes in children with CP. Importantly this study also highlighted differences in immune cell types between neonates with brain injury versus age-matched controls and children with CP, concluding that the immune system may be primed after earlier insult and immune cell changes may persist into school-age.

#### 3.3.4. Genetic Changes and Gene Expression in Case-Controls

One study utilized quantitative PCR to define inherited cytokine SNPs in a moderately sized (n = 188) cohort of infants with and without CP under three years of age. *IL-10* SNPs including rs3024490 and rs1800871 were found to be significantly higher in individuals with CP compared to controls [25]. The study concluded that the *IL-10* SNPs are strongly related to CP and can affect the expression and secretion of the IL-10 cytokine. 

### 3.4. Additional Findings from Single Arm Studies

Two single arm studies were not presented in Table 1 as they did not have a control group [20,23]. Both studies indicated significant differences/associations in inflammatory biomarkers within their CP cohort. One report [23] identified that total white blood cell counts, neutrophil counts and neutrophil-to-lymphocyte ratio were significantly higher in those with CP living in a rehabilitation center setting than those living at home. The other study [20] also showed that inflammatory analytes such as TNF, IL-6 and IL-10 are positively associated with neurotransmitters and neuropeptides (including agoutirelated peptide and Substance P) within a CP cohort via ELISA analysis of CSF.

### 3.5. Subgroup Findings of Biomakers in CP across Included Studies

Limited data was reported across five studies [20,22,23,24,25] that indicated differences in subgroups of those with CP related to age, CP type, severity, topography and etiology (gestational age). A summary of available findings is reported in Table 2. 

#### 3.5.1. Participant Age Subgroup Analysis

One study [24] reported that plasma TNF was significantly increased in younger participants with CP aged 1–3 years compared to those aged 4–12 years with CP. Another study [22], however, detected differences between adults with CP versus children without CP: TGFβ1 and CRP were significantly higher in control children compared to adults with CP. These findings may suggest that participant age may influence the degree of inflammation.

#### 3.5.2. Type and Topography of CP Subgroup Analysis

One study [25] reported that a significant increase in anti-inflammatory IL-10 was found in plasma of the CP group with spastic quadriplegia compared to controls, but not for other CP sub-types. This study also found that frequencies of *IL-10* SNPs of those with spastic quadriplegia versus controls were different for rs1554286, rs151811, rs3024490, rs1800871, and rs1800896. These study results may indicate that the differences in type and topography of CP influence *IL-10* SNPs compared to controls and may represent overall differences in IL-10 production.

#### 3.5.3. Severity of CP Subgroup Analysis

One study found no significant correlations between plasma TNF and CP severity level on the Gross Motor Function Classification System (GMFCS) when looking across the whole group of those with CP [24]. However, significantly increased plasma TNF levels correlated with more severe GMFCS levels in the subgroups of patients with spastic diplegia or quadriplegia as well as those just with spastic diplegia. Moreover, in children with CP in a single arm study [23], neutrophil-to-lymphocyte ratio was significantly increased in children with severe motor impairment (GMFCS IV–V) compared to milder motor impairment (GMFCS II–III). Whilst results are variable, these findings suggest CP severity may be associated with altered inflammatory TNF and neutrophil-to-lymphocyte ratio, particularly in those with more severe CP. 

#### 3.5.4. Etiology of CP Subgroup Analysis

One single-arm study [20] specifically aimed to elucidate the relationship between CSF inflammatory markers and CP etiology, specifically in relation to gestational age. When comparing those with CP born preterm and extremely preterm, there was a significant difference in the correlations of expression between TNF and substance P. Overall, there were 14 unique positive analyte correlations within the preterm birth subgroup that were not found in other term and extremely preterm gestational age subgroups, however only four of these were related to inflammation. Additionally, another 24 unique positive analyte correlations were identified for those born extremely preterm compared to term and preterm, with seven directly related to inflammatory analytes. Whilst evidence is limited, these results suggest that gestational age at birth is implicated in greater expression of inflammatory CSF cytokines and neuropeptides. 

## 4. Discussion

### 4.1. The Evidence of Persistent Inflammation in CP

There is ample evidence to support the role of inflammation in the pathogenesis of brain injury and its detrimental role in neurodevelopment. Preclinical and clinical research has demonstrated that inflammation prevents endogenous brain repair and regeneration following injury [9,30] and persistent inflammation has been postulated to predispose people with CP to further cognitive dysfunction and brain injury [10]. Specifically, published research supports that aberrant glial activation contributes to ongoing injurious brain processes, with advances in neuroimaging supporting this hypothesis [10]. Our findings from this scoping review indicate that some biomarkers of inflammation are altered in people with CP. This has been investigated mostly via alterations in systemic inflammation, commonly assayed from peripheral blood serum and plasma. Changes were detected across markers of inflammatory status measured via cytokine analysis, immune function or genetic changes. Twelve published studies (10 controlled) demonstrate significant changes or associations in one or more inflammatory biomarkers compared to a relevant comparator or within subgroups of those with CP. The most commonly reported changes in inflammation in CP were noted for IL-6, TNF and IL-10. This is consistent with the current literature in infants, with previous systematic reviews indicating that higher circulating levels of cytokines including TNF and IL-6 are associated with abnormal neurological findings, including CP [4]. 

Importantly, our scoping review highlights that inflammation can persist in CP well beyond the acute brain injury period, with significantly higher systemic inflammation from childhood, through to adolescence compared to relevant controls. Some differences in gene function were found in participant groups aged up to 18 years. We also note that some studies report differences in inflammatory status related to age; two studies showed that younger children have greater systemic inflammation compared to older children and adults. Interestingly, in the one study that identified no significant inflammatory differences in plasma CRP between those with CP and controls, all participants were adults. This may indicate that inflammatory status and immune function change over time and may become less pronounced with increasing age, however more research is required to elucidate this hypothesis. 

Whilst there are several common biomarkers under investigation including IL-6, TNF and IL-10, there remains high heterogeneity between studies. A total of 30 biomarkers of inflammation as well as a number of additional genes were examined and studies included varied sample types, analysis methods, controls, and age ranges and presentations of participants. Even across the most commonly investigated cytokines of IL-6, TNF and IL-10, studies reported mixed significant and non-significant findings. Remarkably, the majority of studies conclude that there are differences in inflammation in CP, spanning more than 38 significant findings. These primarily include alterations in systemic inflammatory and immune function, and more research should be prioritized to investigate these changes in more detail and in larger cohorts with harmonized biomarker panels. 

### 4.2. Differences in Persistent Inflammation in CP and within Subgroups

CP is a multifactorial and heterogeneous condition, stemming from diverse etiologies and patterns of brain injury, with varied severity and subtype [31]. As such, we cannot assume that inflammation between two individuals with CP will be the same. Whilst data in this area is still emerging, the findings of our scoping review suggest that the variability between and within studies might be explained by subgroups of people with CP. Specifically, we note that participant age, severity of CP, type/topography and etiology may be important to consider when assessing inflammation. We present that those with more severe CP may have higher levels of inflammation, as well as those with spastic quadriplegia. One included study also supports that those with CP born preterm have distinct inflammatory biomarkers, complementing the previously established literature showing unique biomarker profiles in those born preterm and the risk of developing CP [4]. Whilst this data may indicate that there are differences in inflammation both in CP and within subgroups, there may also be contributing factors to persistent inflammation that are not controlled for. For instance, in more severe CP, individuals may have reoccurring infections, micro-aspirations and more extensive muscle contractures [32]. These factors could ultimately explain the higher levels of systemic inflammation observed in both inflammatory and anti-inflammatory cytokines [24,25]. Moreover, the one study that found higher plasma IL-10 in those with spastic quadriplegia may suggest that increased muscle tone alters inflammation [25]. However, interpretation of this finding is limited as other types of CP are not commonly studied. There remains a notable bias towards participants with spastic CP, likely due to prevalence of spastic CP over other CP types. Additionally, we noted that most included studies of this review had a modest sample size (n ≤ 150 total) with a mixed participant demographic. Only one study had a large sample size of n = 479 [25] in order to analyze genetic polymorphisms with subgroup analyses of CP subtypes. Future studies will require larger sample sizes to ensure that further subgroup analyses can be conducted, are adequately powered to detect differences and control for subgroup and comorbidity diversity, especially in this heterogenous and complex condition. 

### 4.3. Targeting Inflammation as a Comorbidity of CP

It has been previously proposed that understanding “persistent inflammatory mechanism could lead to safe and effective therapies to treat brains that have experienced developmental disruption long after the initial insult” [9]. This is an interesting concept, and our scoping review demonstrates that inflammation should be recognized as a comorbidity of CP, that is, a factor that coexists alongside the movement and postural impairments experienced by people with CP. Given the remaining uncertainties highlighted above, more research to improve our understanding of the extent of inflammation, its mechanisms in CP, as well as who are most likely to have inflammation and require treatment, will be important next-steps.

Importantly, whilst this review has highlighted a number of inflammatory biomarkers of interest (e.g., IL-6, TNF and IL-10), it is unlikely that any one given cytokine alone will be implicated in the pathogenesis and long-term inflammatory status associated with CP. Instead, a broad immunomodulatory and anti-inflammatory strategy may have its place and are now under investigation in CP. For instance, cell therapies including umbilical cord blood have been demonstrated to improve motor function in children with CP [33,34] primarily working via immunomodulation. Results from this scoping review support the hypothesis that there is persistent inflammation to be targeted. Uncovering the role, extent and impact of this inflammation may also enable research into more treatments that target inflammation. 

Alternatively, given the variability in inflammatory biomarkers and likely heterogeneity in any given person with CP, personalized medicine may also be appropriate approach for targeting inflammation in this condition. For instance, we have demonstrated that levels of systemic inflammatory TNF are higher in three studies with a CP sample of n = 117. If this finding is confirmed in more participants with CP compared to controls, TNF may prove a future target for personalized medicine. There are now success stories from clinical translation of inflammatory cytokine drug targets for neonatal conditions including white matter injury and bronchopulmonary dysplasia. These include the use of IL-1 receptor antagonist (anakinra) [35,36] and IL-37 as an endogenous regulator of inflammation that broadly suppresses innate and adaptive immunity [37,38]. A similar approach may be developed in CP if researchers can establish the importance of any one given cytokine for the condition. Additionally, identifying those with persistent inflammation may also help us to understand responders and non-responders to treatments with an inflammatory mechanism of action.

### 4.4. Outstanding Unknowns of Inflammation in CP

This review highlights that inflammation has a number of dimensions; there are multiple contributors to inflammation spanning the transcriptome that contains the genome, proteome, metabolome, leading to phenotypic changes [39]. However, the majority of results in this scoping review are only reflective of changes at the protein level mainly from peripheral blood serum and plasma. To continue establishing inflammation as a comorbidity of CP, more research should be done to elucidate the contribution of upstream and downstream inflammatory factors. This may also help to uncover novel targets. Additionally, the origins of inflammation associated with early brain injury and resulting CP are yet to be fully understood. For instance, *is inflammation programmed during the brain injury? If this inflammation remains sustained, why and how*? The causes of inflammation may be cellular or from epigenetic programming from the initiating insult [13]. Not to mention, the confounding role of aberrant inflammation following perinatal infection and neonatal sepsis in CP [40,41], as well as maternal immune activation [42] and congenital abnormalities [43]. Current evidence also suggests that more than 30% of all CP may have a genetic cause, with four main types of DNA variations contributing to the pathogenesis of the condition [7]. Genetic and epigenetic changes may not only increase the risk of developing CP, but may also have a role in ongoing signaling pathway dysregulation like Wnt and glycogen synthase kinase-3. These signaling pathways are critical for brain development and neurogenesis in early life but also support regeneration, synaptic plasticity and homeostasis in adults. Information regarding these changes in CP, such as how inflammation is sustained and the implications of persistent inflammation are only just becoming apparent from research findings.

As we work to comprehend how inflammation is sustained, it is also important to understand whether blood biomarkers of systemic inflammation (as commonly reported in this scoping review), remain a direct indicator of central nervous system inflammation in established CP. Whilst we saw one study that analyzed CSF, more should be done to contrast and compare systemic and central inflammation. Research into neurological conditions including stroke has demonstrated that there is strong peripheral immune-brain crosstalk following injury [44,45]. This crosstalk can be bidirectional and therefore systemic immune responses to stimuli may exacerbate brain inflammation, and vice versa. Not to mention, this review has primarily focused on the detrimental effects of inflammation, however the protective and reparative functions of inflammation in the setting of CP have yet to be considered.

This review also highlights that different causal pathways and etiologies of CP, as well as subtypes, may contribute to persistent inflammation. However, evidence is still emerging and most studies did not provide details of CP etiology or investigate subgroup analyses of participants in relation to inflammation. As stated above, powering for this type of investigation is necessary but will be challenging. 

### 4.5. Limitations

We acknowledge some limitations of this study, primarily the heterogeneity of inflammatory biomarkers that restricted synthesis as well as prevented clear directionality findings of any one given cytokine. This limits the clinical utility of the results and we cannot conclude with certainty what inflammatory biomarkers may contribute as a pathogenic mediator or comorbidity of CP. Inflammatory biomarker heterogeneity also meant that analytes were not discussed in the context of pro- and anti-inflammation. The scoping review was also limited by the ability to adequately define the CP cohort, with only five studies reporting CP etiology and most not reporting all participant features like type, topography and severity of CP. As different causal pathways and presentations of CP may influence inflammatory burden and persistence, adequate reporting of participant demographics remains imperative. Additionally, a number of studies included in this scoping review present results in reference to a range of different comparators. In one instance, children with CP were compared to adult controls and this may have influenced results. Overall, the heterogeneity of the participants, biomarkers studied and methodologies employed, in addition to failures in controlling for factors that may influence results (adequate controls, subgroups of CP), limits the scoping review findings and generalizability to people with CP. 

## 5. Conclusions

This scoping review is the first to present the summarized research to date on inflammation in CP. The available evidence indicates that inflammation is pathogenic in CP and may persist in various forms including immune and genetic changes, years after the original injury. Given that persistent inflammation in CP may have deleterious effects across development and into adulthood, inflammation should be recognised as a comorbidity of CP. Whilst there are a number of common cytokines under investigation including IL-6, TNF and IL-10, there remains high heterogeneity between studies and more research is required to improve the strength of the evidence. It is still unclear why inflammation persists, how it persists and if there are subgroups of people with CP who may have more extensive inflammation. Efforts should be made to investigate inflammation in CP in well-controlled, larger studies that adequately address population heterogeneity, which may in turn help support future research into novel treatment options. 

## Figures and Tables

**Figure 1 jcm-11-07368-f001:**
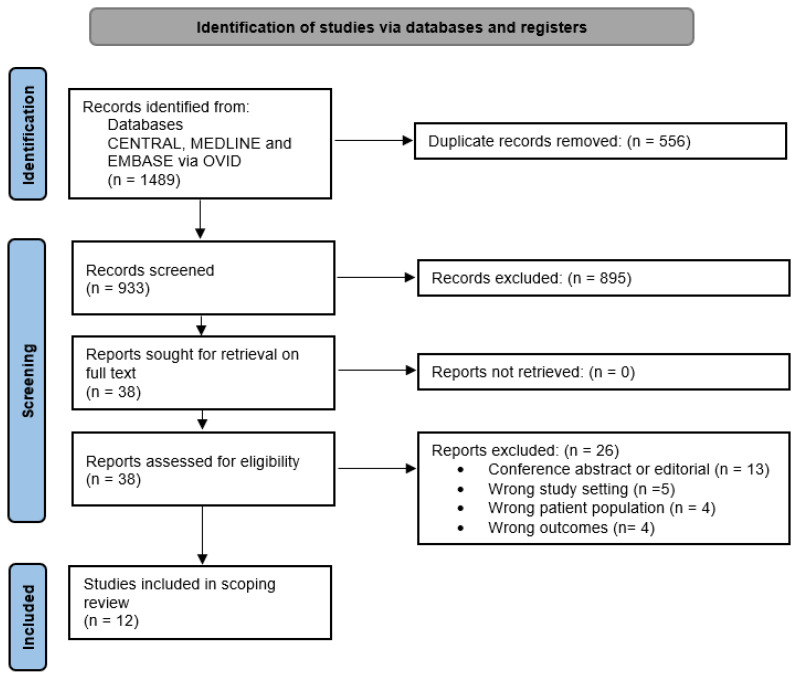
PRISMA Flow Diagram.

**Table 1 jcm-11-07368-t001:** Study characteristics and main findings.

Citation	CP Group/s and Comparator Details	Etiology of CP Group	*Result Type* Sample Type Method of Analysis: Inflammatory Markers	Reported Significant Findings Related to CP
Tao 2008 [19]	Children with CP: Type/topography/severity: not reported Mean age 5.2 years, range 0–10 (n = 31, 55% male) Neonatal controls with asphyxia or infection: Mean age 6.3 days, range 0–10 (n = 37, 54% male) Age-matched controls, no CP: Mean age 4.8 years, range 0–10 (n = 40, 58% male)	HIE and/or infection	*Cytokine analysis* Serum ELISA: TNF, IL-6	↑ TNF in CP vs. neonatal controls ↑ TNF in CP vs. age-matched controls ↑ IL-6 in CP vs. age-matched controls
Lin 2010 [5]	Children and adolescents with CP: Spastic/di, tri or quad/GMFCS II-V: Mean age (SD) 7.2 (±3.6) years (n = 32, 59% male) GA-matched controls, no CP: Mean age (SD) 6.2 (±2.2) years (n = 32, 44% male)	Preterm with PVL	*Cytokine analysis* Plasma ELISA: TNF, IL-6	↑ TNF in CP vs. GA-matched controls
			*Immune function* Supernatant from PBMCs, +/− LPS ELISA: TNF, IL-6 Flow cytometry: TNF	↑ TNF (ELISA) after LPS in CP vs. GA-matched controls ↑ TNF (flow cytometry) before vs. after LPS in CP group
			*Immune function* RNA from PBMCs, +/− LPS RT-PCR: 84 TNF genes (TNF ligand/receptor signalling)	↑ 10 genes (*CAD, CASP2, CRADD, EDA2R, IKBKG, TAK1/TGF1a, JNK/JNK1, 4-1BB/CD137, TL1/TL1A, TRAF3)* in CP vs. GA-matched controls
Koh 2015 [21]	Children with CP: Type/topography/severity: not reported Mean age not reported (n = 14, sex not reported) Adult controls, no CP Mean age not reported (n = 14, sex not reported) Cord blood from healthy neonates (n = 14, sex not reported)	Not reported	*Immune function* Mobilised PBMCs, +LPS/PMA and ionomycin Flow cytometry: TNF, IL-1β, IL-2, IL-3, IL-6, IL-8, IL-9, GM-CSF	↓ IL-1β in CP vs. adult controls ↑ GM-CSF in CP vs. adult controls ↑ IL-8 in CP vs. adult controls
Wu 2015 [24]	Children with CP: Spastic/tri, quad, di, mono, hemi/GMFCS I-V Mean age (SD) 3.7 (±2.3) years (n = 54, 59% male) Age-matched controls, no CP: Mean age (SD) 4.6 (±3.1) years (n = 28, 54% male)	Not reported	*Cytokine analysis* Plasma ELISA: TNF	↑ TNF in CP vs. age matched-control
Von Walden 2018 [26]	Children and adolescents with CP: Spastic/ topography not reported/GMFCS I-II, IV-V Mean age 15.5 years, range 9–18 (n = 18, another n = 2 with ABI, 85% male) Children and young adult controls, no CP: Mean age 15.1 years, range 7–21 (n = 10, 80% male)	Not reported	*Cytokine analysis* Skeletal muscle biopsy RT-PCR: *IL-1β, IL-1R, IL-6, IL-6R, TNF, TWEAK, IL-8, IL-10*	↑ *IL-1β* in CP vs. controls ↑ *IL-6* in CP vs. controls ↑ *TNF* in CP vs. controls
Xia 2018 [25]	Infants and children with CP: Spastic and non-spastic/hemi, di, quad /GMFCS not reported Mean age for genotyping/plasma collection (SD) 16.2 (±12.7) months; 20.8 (±14.4) months (n = 282, 65% male) Infant and children controls, no CP: Mean (SD) age for genotyping/plasma collection 24.0 (±16.4) months; 21.6 (±13.8) months (n = 197, 77% male)	Mixed: HIE, PVL, other/not specified	*Cytokine analysis* Plasma ^1^ ELISA: IL-10	↑ IL-10 in CP vs. controls
Pingel 2019 [22]	Children and adolescents with CP: Type not reported/hemi, di, quad/GMFCS I-V: Mean age (SEM) 10.31 (±1.1) years (n = 14, 50% male) Adults with CP: 38.8 (±3.6) years (n = 10, 60% male) Adult controls, no CP: Mean age (SEM) 36.53 (±3.8) years (n = 10, sex not reported)	Not reported	*Cytokine analysis* Plasma ELISA: TGF*β*1, CRP, IL-6	↑ TGFβ1 and CRP in children with CP vs. adult controls ^2^
Ng 2021 [18]	Adults with CP: Type/topography/severity: not reported Mean age (SD) 25 (±5.39) years (n = 64, 55% male) Older adults’ with mild cognitive impairment, no CP (mean age (SD) 66.95 (±4.29) years) (n = 40, 30% males) Older adults’ controls, no MCI, no CP: (mean age (SD) 71.8 (±5.66) years) (n = 56, 20% males)	Not reported	*Cytokine analysis* Plasma ELISA: CRP	None
Taher 2021 [6]	Children and adolescents with CP: Type/topography/severity: not reported Mean age: “School-aged children”, age not reported (n = 10, 80% male) School aged children post NE, no CP (n = 10, 70% male) Mean age: not reported School aged children no NE, no CP (n = 23, 78% male) Mean age: not reported Neonates with NE (n = 30, 50% male) Mean age: not reported Neonates, no NE (n = 17, 53% male) Mean age: not reported	NE	*Proportions of immune cell types* Whole blood Flow cytometry: T cells (CD3+), B cells (CDCD3- CD19+), NK cells (CD3-/CD56+), MAIT cells (CD3+/Va7.2+/CD161+), iNKT cells (CD3+/Va24Ja18+) Vδ1 TCRs (CD3+/ Vδ1+), Vδ 2 TCRs (CD3+/ Vδ2+)	↑ T-cells (absolute and % freq) in children with CP vs. school-aged children ↑ Vδ2 T cells (absolute and % freq) in children with CP vs. school-aged children ↑ iNKT cells (absolute and % freq) in children with CP vs. school-aged children ↑ CD4− CD8− T cell frequencies in children with CP vs. school aged children ↓ Vδ1 T cells (absolute and % freq) in children with CP vs. school aged children ↓ MAIT cell (% freq) in children with CP vs. school aged children ↓ B cell (% freq) in children with CP vs. school aged children
			*Cytokine analysis* Serum ELISA: IFN-y, TNF, IL-2, IL-5, IL-6 IL-8, IL-9, IL-10, IL-15, IL-17A, IL-21, IL-22, IL-23	↓ IL-8 in children with CP vs. school aged children post NE
Zareen 2010 [9]	Children and adolescents with CP: Type not reported/topography not reported/GMFCS II, III, V Mean age (SD) 10.08 (±1.67) years, range 1–16 (n = 12, 67% male) Age-matched controls, no CP Mean age (SD) 9.08 (±1.08) years, range 6–14 (n = 12, 67% male)	Mixed: NE, congenital infection and stroke	*Cytokine analysis* Serum ELISA: IL-1α, IL-1β, IL-6, IL-8, IL-10 IL-18, IL-1Ra, TNF INF-γ, GM-CSF	None
			*Immune function* Supernatant from whole blood + LPS ELISA IL-1α, IL-1β, IL-6, IL-8, IL-10, IL-18, IL-1Ra, TNF, IFN-γ, GM-CSF	↑ IL-1α, IL-1β, IL-2, IL-6 after stim in CP vs. age-matched controls

Significance is defined as reported in the included papers (*p* < 0.05). “Infants” (<1 year); “children” (1–9 years); “adolescents” (10–19 years); “adults” (20–59 years), “older adults” (60+ years). ^1^ Whole blood DNA Genotyping SNPs excluded from table as results were not relevant. ^2^ Additional significant finding detected comparing children with CP to adults with CP. Results appear as a subgroup finding in Table 2 related to age. ↑ increase; ↓ decrease; ABI, acquired brain injury; CD, cluster of differentiation; CP, cerebral palsy; CRP, C-reactive protein; CSF, cerebrospinal fluid; ELISA, enzyme-linked immunosorbent assay; di, diplegia; FACS, fluorescence activated cell sorting; freq, frequency; G-CSF, granulocyte-colony stimulating factor; GMFCS, gross motor function classification system; HIE, hypoxic ischemic encephalopathy; ID, intellectual disability; IKK-γ, inhibitor of nuclear factor kappa-B kinase subunit γ; hemi, hemiplegia; IFN, interferon; IL, interleukin; iNKT, Invariant natural killer T; IP, interferon γ-induced protein; JNK, c-Jun N-terminal kinases; LPS, lipopolysaccharide; MAIT, Mucosal-associated invariant T; MIP1β, macrophage inflammatory protein; MCP, monocyte chemotactic protein; n, sample; NE, neonatal encephalopathy; NLR, neutrophil to lymphocyte ratio; PBMCs, peripheral blood mononuclear cells; PMA, phorbol 12-myristate 13-acetate; PROM, premature rupture of membranes; RA, receptor antagonist; RANTES, regulated on activation normal T expressed and secreted; RNA, ribonucleic acid; RT-PCR, reverse transcription polymerase chain reaction; SD, standard deviation; SEM, standard error of the mean; SNP, Single nucleotide polymorphism; TNF, tumor necrosis factor; TLR, Toll-like receptor; TCR, T-cell receptor; TAK, transforming growth factor-β-activated kinase; TWEAK, tumor necrosis factor-like weak inducer of apoptosis; quad, quadriplegia; vs., versus.

**Table 2 jcm-11-07368-t002:** Main findings related to subgroups of cerebral palsy.

Theme	Reported Significant Findings in Subgroups of Those with CP	Citation
Age	↑ plasma TNF in CP vs. younger (1–3 years) and older (4–12 years) controls ↑ plasma TNF in younger subjects (1–3 years) with CP vs. older subjects (4–12 years) with CP ↑ TGFβ1 and CRP in children with CP vs. adults with CP	Wu 2015 [24] Pingel 2010 [22]
Type/Topography	↑ plasma IL-10 in the CP group with spastic quadraplegia vs. controls Frequencies of allele and genotype those with spastic quadraplegia vs. controls of *IL-10* polymorphisms: rs1554286, rs151811, rs3024490, rs1800871, and rs1800896	Xia 2018 [25]
Severity	↑ plasma TNF levels correlated with higher GMFCS in spastic diplegia or quadriplegia, and spastic diplegia	Wu 2015 [24]
	↑ whole blood NLR in children with severe motor impairment (GMFCS IV-V) vs. mild motor impairment (GMFCS II-III) living in a rehabilitation centre ↑ whole blood NLR positively correlated with higher GMFCS level	Riewruja 2020 [23]
Aetiology	4 unique discrete correlations in CSF related to inflammation, specific to preterm birth vs. term and extremely preterm: IL-10 and MIP-1β; IL-12p70 and TNF; IL-1Ra and IL-10; IL-1Ra and MIP-1β. 7 unique discrete correlations in CSF related to inflammation, specific to extreme preterm birth vs. term and preterm birth: IP10 and MIP-1β; IL-1α and IL-6; IL-1α and IL-10; IL-1α and RANTES; IL6 and IL10; IL-6 and RANTES; IL-10 and RANTES	Gorack-Postle 2021 [20]

CP, cerebral palsy; CRP, C-reactive protein; CSF, cerebrospinal fluid; GMFCS, gross motor function classification system; IL, interleukin; MIP-1β, macrophage inflammatory protein; NLR, neutrophil to lymphocyte ratio; Ra, receptor antagonist; RANTES, regulated on activation normal T expressed and secreted; TNF, tumor necrosis factor; vs., versus.

## Data Availability

Data is contained within the article or Appendix A.

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
