# Peer review of "Persistent Inflammation in Cerebral Palsy: Pathogenic Mediator or Comorbidity? A Scoping Review"

_jcm, 2022, doi:10.3390/jcm11247368_

Round 1

Reviewer 1 Report

The scoping review is good, but it requires some minor corrections.

1.       Is the scoping review id registered in PROSPERO?

2.       There is variability in participants features, so the results showing presence of inflammatory biomarkers needs to be clearly mentioned representing each feature.

3.       The etiology of CP in relation to inflammatory markers needed to be elaborated

4.       Is there any association between inflammatory biomarkers and Wnt, GSK 3 signalling pathways in cerebral palsy? If yes, please mention.

5.       Is there any exclusion criteria in the selection of participant feature? If yes, please mention

Author Response

We thank you for your review. Please see the attachment for our detailed response.

Reviewer 2 Report

CP is a complex problem and, as it turns out, not fully understood. The work is interesting, it presents the causes of CP from a different perspective. Recently, it has been found that most patients combine two factors: premature birth and difficult delivery with neonatal asphyxia (or hypoxia). It is not known why the birth takes place prematurely. There are certainly many reasons for this, some of which we know and some of which we do not.

First, please provide the current definition of CP.

The concept of an inflammatory cause of CP is interesting. However, it is not explained why there is inflammation, what happens during pregnancy, maybe the mother is sick, maybe there is some congenital defect that causes inflammation. I propose to develop this aspect in the discussion. Also, if the anti-inflammatory IL-10 was found in the plasma of the spastic tetraplegia CP group compared to the control group and not in the other forms of CP, then the spastic form causes inflammation. This is logical because increased muscle tone can trigger an increase in some components in blood tests. We do not routinely test immunological indicators. Might be worth doing some research on this.

However, significantly increased plasma TNF levels correlated with more severe GMFCS levels in subgroups of patients with spastic or quadriplegic CP. It is worth discussing why this is only the case in spastic forms and not in other forms.

In more severe CP, patients may have recurrent infections, microaspirations, and more extensive muscle contractures. These

factors may ultimately explain the higher levels of systemic inflammation often seen. Most of the studies included in this review had a modest sample size (n=<150 total) with mixed participant demographics - please discuss this in the available articles

Author Response

(The authors gave the same response as above.)

Round 2

Reviewer 2 Report

Article revised as recommended. It can be published in this form.